# Competitiveness of Food Industry in the Era of Digital Transformation towards Agriculture 4.0

Ismael Cristofer Baierle [1,*], Francisco Tardelli da Silva [1], Ricardo Gonçalves de Faria Correa [1], Jones Luís Schaefer [2], Matheus Becker Da Costa [3], Guilherme Brittes Benitez [4] and Elpidio Oscar Benitez Nara [4]

1 Graduate Program in Agro-Industrial Systems and Processes, Federal University of Rio Grande, Rio Grande 96203-900, Brazil
2 Department of Production Engineering, Federal University of Santa Maria (UFSM), Santa Maria 97105-900, Brazil
3 Department of Industrial Systems and Processes, University of Santa Cruz do Sul (UNISC), Santa Cruz do Sul 96815-900, Brazil
4 Polytechnic School, Pontifical Catholic University of Parana (PUCPR), Londrina 86072-360, Brazil
* Correspondence: ismaelbaierle@hotmail.com

**Abstract:** Industry 4.0 and its technologies can potentially increase business competitiveness in the age of digital transformation through the implementation of its technologies. These digital technologies are increasingly present in the food industry, characterizing the concept of Agriculture 4.0. This digital transformation is a reality; however, it is unclear which digital technologies are most useful for each industry sector. Therefore, this paper seeks to explore the current state of implementation of digital technologies in different industrial sectors and which digital technologies should be leveraged to increase the performance of the agribusiness system. To do so, we used secondary data from a large-scale survey of 28 industrial sectors, representing 2225 companies in the Brazilian industry. Analyzing the different industrial sectors allowed us to present a framework of digital transformation to boost food industry competitiveness towards Agriculture 4.0. The results show that the food industry usually uses only one digital technology, showing the need for simultaneous and joint investments in the other technologies presented in this research. Public policies must be directed to encourage the expansion of digital technologies in the food industry.

**Keywords:** Industry 4.0; Agriculture 4.0; competitiveness; open innovation; emerging countries; digital transformation; agroindustry

## 1. Introduction

The advance in the competitiveness of industries passing through the new industrial revolution, the digitalization of manufacturing, called Industry 4.0, is driven by groups of emerging and disruptive technologies [1]. Through technological advances, there have been significant increases in industrial productivity since the first industrial revolution until where we are now, in the midst of a new industrial transformation driven by some key technologies such as big data, the cloud, and IoT [2], bringing important contributions in terms of scalability and interoperability of solutions [3]. The development of advanced electronic, information, and manufacturing technologies is changing the production process of companies [4], which transforms traditional manufacturing into intelligent manufacturing, increasing the competitiveness and flexibility of organizations [5,6]. It is proven that the process of Industry 4.0 and digital transformation in emerging countries has its peculiarities and differs from developed countries [7]. Consequently, emerging countries have their perceptions and particularities of Industry 4.0 technologies [8]. Digital transformation and innovation processes are also making their way into the food industry, giving rise to Agriculture 4.0 [2,9].

The food industry encompasses different players, from farmers to food manufacturing and processing companies. As the world population grows [10,11], one of the challenges for agriculture and the food industry is to increase or optimize production and processing [12,13] sustainably. Agriculture plays a very important role in many countries' gross domestic product (GDP) [14]. In the world, agriculture accounts for 6.4% of GDP, and in some countries, it is the dominant sector [15]. In emerging countries such as Brazil, agriculture is critical to improving economic performance in the coming years [16,17]. In countries with more developed economies, agriculture can also be used to leverage their international market share of agricultural products [18]. In this context, digital transformation is emerging as an enabler for agricultural development and the food industry, transforming traditional food systems into advanced, technology-based systems [19,20].

Through Industry 4.0 technologies and digital transformation, conventional agriculture and the food industry will give rise to Agriculture 4.0 [20,21]. Digital transformation is defined as a process of change in the use of digital technologies that generates better performance in the processes of a business [22]. Digital transformation is important because it stimulates the industry to seek changes [23–26]. On the other hand, digital transformation can increase uncertainty, as managers need to know what to commit to and what needs to be adapted faster [27–29], generating results faster. Although digital transformation is becoming a strategic imperative in traditional industry sectors [30,31], how these industries will digitally transform remains unclear. There is a need to understand digital transformation and identify which digital technologies are most suitable for each industry sector [32].

Moreover, the impacts of digital transformation can be measured to help verify improvements in competitiveness. The measurement of competitiveness can be used as a tool for strategic management, making it possible to monitor and optimize a company's performance [33,34]. From this measurement, companies in the food industry can make comparisons. To improve their competitiveness, they must correctly invest their resources, adapt to the market, manage knowledge, and integrate new technologies [35]. Still, the uncertainties of which Industry 4.0 digital technologies should be adopted in each industrial sector concern managers, making it difficult to invest in such technologies, especially in the food industry. The uncertainties worldwide affect all industrial sectors and are the main barrier to going forward with Industry 4.0/Agriculture 4.0, especially in emerging countries. Emerging countries do not have the economic empowerment and technical knowledge to bet in an environment with many uncertainties [36,37]. Measuring digital technologies' impact on competitiveness might be an approach to guide industries to advance in their journey towards development.

## 1.1. Digital Technologies Adoption in Industrial Sectors

To advance in Agriculture 4.0, digital technologies from previous stages are required [20]. As stated by the CNI report [38] and Refs. [39,40], technologies such as sensors, CAD-CAM systems, and MES-SCADA systems from the third industrial revolution are still fundamental to Industry 4.0. It is important to highlight this due to the need for a minimum technological architecture to implement the integration and digitization concepts from Industry 4.0 [2]. Based on this, Table 1 presents a list of eleven technologies frequently associated with the Industry 4.0 concept [38,41,42].

The digital technologies presented in Table 1 are directly related to digital transformation and were used in the CNI survey; they comprise concepts such as vertical integration, horizontal integration, and end-to-end engineering [24,40]. Vertical integration refers to integrating all elements at the factory level until management-level decision-making [43]. Horizontal integration refers to collaborating with different actors (e.g., suppliers and manufacturing enterprises) in the value chain, exchanging information and resources in real-time [44]. End-to-end engineering integrates the entire value chain of the product, from its raw material until after-sales, to optimize the product and industrial processes [45].

**Table 1.** Digital technologies of Industry 4.0.

| Digital Technologies | ID |
| --- | --- |
| Computer-aided manufacturing projects CAD/CAM | CAD-CAM |
| Integrated engineering systems for product development and product manufacturing | IES |
| Digital automation without sensors | DAwS |
| Digital automation with process control sensors | DAS |
| Digital automation with sensors for product and operating condition identification, flexible lines | Flex |
| Remote monitoring and control of production through systems such as MES and SCADA | MES-SCADA |
| Simulations/analysis of virtual models (finite element, computational fluid dynamic, etc.) for design and commissioning | Simulation |
| Collection, processing, and analysis of large quantities of data (big data) | Big Data |
| Incorporation of digital services into products ("Internet of Things" or product service systems) | IoT-PSS |
| Additive manufacturing, rapid prototyping, or 3D printing | 3D |
| Use of cloud services associated with the product | Cloud |

These three integrations are fundamental to achieving better results in industrial performance in the fourth industrial revolution [46,47]. However, which digital technologies should be integrated to achieve these results is a concern to managers. The adoption of Industry 4.0 digital technologies depends on the context of each industrial sector [39]. Moreover, this adoption also depends on the country's context [48]. Despite the industrial sectors having similar manufacturing and process characteristics apart from the country, the context of each country should be considered, especially when related to their R&D investment level and technology acquisition. Therefore, the competitiveness and technology adoption levels will differ from sector to sector [49].

The use of digital technologies arising mainly from Industry 4.0 has been the subject of research for over a decade, but there are still questions about how to apply them in practice. In this sense, this article seeks to explore the current status of implementation and knowledge of the leading technologies in different industry sectors in an emerging country, focused on the food industry, which is one of the links that make up the Agriculture 4.0 concept. To achieve the goal, two questions arise: (i) What is the current status of digital technology implementation in different industry sectors? (ii) Which digital technologies can be leveraged to increase the food industry's performance towards Agriculture 4.0? The research questions take into account that different industry sectors have particularities. Still, it is important to analyze all sectors, seeking to understand how one sector can contribute to others [39], and especially how other sectors can contribute to the digital transformation of the food industry, an integral part of Agriculture 4.0.

These questions were answered by ranking the competitiveness of different industrial sectors when implementing digital technologies to understand into which level the food industry fits. In answering the research questions, we present a framework of digital transformation to boost food industry competitiveness towards Agriculture 4.0. To do this framework, we analyze secondary data from a large-scale survey applied in Brazil by the National Confederation of Industries (CNI), comprising a sample of 2225 companies from different industrial segments in this emerging country. We used the multiobjective optimization by ratio analysis (MOORA) method, considering two weighting emphases for the criteria. The first emphasizes the alternatives and criteria with different weights, while the second considers the Fuzzy Delphi method's integration for the weights' unification. These procedures allowed us to provide an overview of the most prominent digital technologies for each industrial sector, always focusing on the food industry and giving an initial perspective to managers in an emerging country like Brazil. Finally, this study provides insights into how the 28 industrial sectors can contribute to the advancement of digital transformation in the food industry, contributing to developing the concept of Agriculture 4.0.

The results show that the most developed industries regarding digital technologies are the electrical, electronics, plastics, and vehicle industries, while the food industry is one of the industries that uses digital technologies the least. Emerging countries can contribute to developing digital technologies by doing groundwork and preparation, which can be replicated in other countries. Finally, the presented framework shows which technologies

are the greatest drivers of competitiveness in the food industry towards Agriculture 4.0. This information should be the target of future research, showing how it should and can be used in practice to become accessible to any type or size of company. The advancement of research focused on certain technologies and applications contributes to the advancement of digital transformation in the food industry and the development of the concept of Agriculture 4.0.

The remainder of the paper is organized as follows. Section 3 details the methodological procedures used, Section 4 presents the results obtained, Section 5 brings a broad discussion about the findings, and Section 6 presents the paper's conclusions.

## 2. Correlated Works

The current literature has provided us with the most important digital technologies responsible for transitioning from classical (current) digitization to Agriculture 4.0. The present work sought to investigate research on how the food industry can contribute to advancing the concept of Agriculture 4.0. Agriculture and the food industry are becoming increasingly innovative with new infrastructures, computing platforms, and biotechnologies such as gene editing or synthetic food production [50]. Digital technologies are changing how companies do business and establish deeper relationships with customers, suppliers, and other stakeholders [51–53]. For example, the food industry comprises many companies and represents the largest contributing industry in the European Union regarding economic output and employment [54,55]. In this context, Brazil is a major food supplier for the world market, which is constantly increasing. The rapid growth and development of digital technologies in the agriculture and food industry will lead to profound modernization in key sectors of the world economy [56]. Despite its importance, studies on the agri-food sector are scarce [57], as is information regarding digital transformation, which is a challenge for companies that were not originally digital and are on the verge of migrating to Agriculture 4.0 [58,59]. Agriculture 4.0 consists of the adoption of digital technologies to manage agricultural or industrial processes [60,61] to monitor different parameters [62] based on a data set [63–65]. The data can power food from crop cultivation to processing [66,67], lowering production costs and eliminating non-essential inputs [68,69]. The authors of [70] report that adopting new digital technologies promotes a sustainable agricultural production chain. Agriculture 4.0 is also defined as the implementation of information and communication technologies (IoT, GPS, big data) on farms by farmers seeking to improve the quality and increase the productivity of their farms [71,72]. For [9], Agriculture 4.0 should follow the examples of the evolution of European industries. The authors of [73] add that Agriculture 4.0 represents an excellent opportunity to consider the variability and uncertainties in the agricultural and food chain. For the authors [74,75], economic development and digital technologies concern the sustainable environment. This evidence of the importance of studies focused on economic development will also bring sustainable development in the whole productive chain in which the food industry is inserted.

## 3. Materials and Methods

The data collection of this research is characterized by the extraction of the answers to a questionnaire conducted by the CNI with Brazilian companies, titled "Special survey on Industry 4.0 in Brazil". The CNI is an agency representing Brazilian industries in 31 industrial sectors with 1250 employers' unions, with approximately 700 thousand industries affiliated [38]. Its main role is supporting companies in issues that impact industrial performance and the country's economy and stimulating research, innovation, and technological development. Figure 1 summarizes our research method in four main steps.

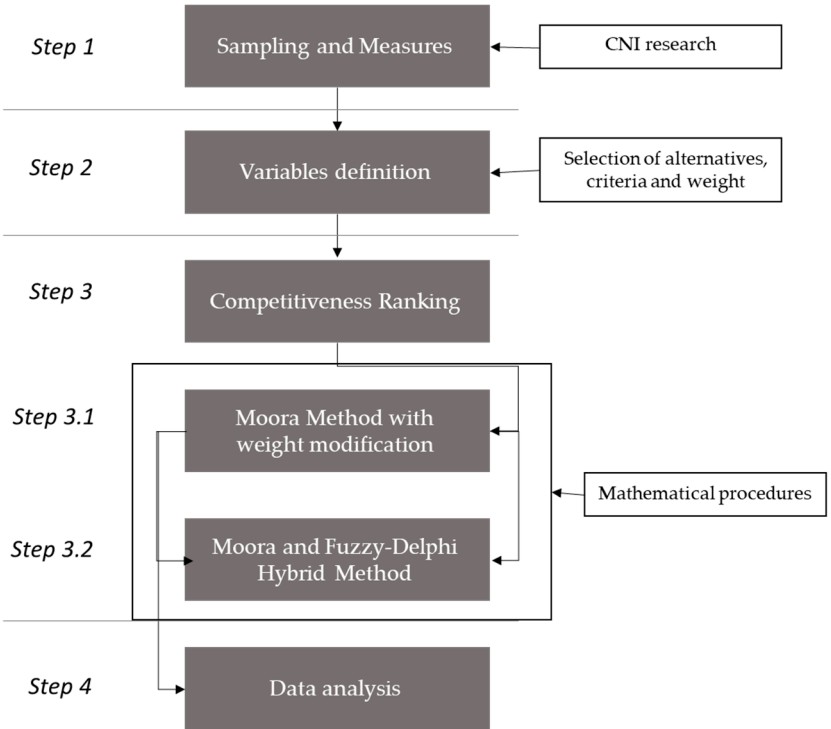

**Figure 1.** Research steps.

### 3.1. Step 1—Sampling and Measures

The CNI questionnaire aimed to identify the situation of Brazilian industries in the context of Industry 4.0 digital technologies [38]. The importance of these technologies to boost competitiveness, the use of these digital technologies, expected benefits, internal and external barriers in implementing the respective digital technologies, and government policies to foster Industry 4.0 in Brazil were evaluated. The questionnaire was sent to 7836 randomly selected companies within the population, comprised of manufacturing-related industries. The total number of responses obtained was 2225, representing a response rate of 28.39%. The sample of companies was composed of 910 small companies (up to 99 employees), 815 medium-sized companies (up to 499 employees), and 500 large companies (over 500 employees).

### 3.2. Step 2—Variables Definition

We used 11 digital technologies from the CNI survey (see Table 1). We utilized acronyms (e.g., 3D for additive manufacturing) to identify Industry 4.0-related digital technologies. Initially, the CNI surveyed 31 industrial sectors, but some sectors did not answer the survey. Therefore, three sectors were withdrawn for not presenting information: "Mining of coal and extraction of petroleum", "Mining support service activities", and "Tobacco products". Table 2 presents the final list of industrial sectors considered in our analysis.

For both research questions, answers related to "None of the items listed", "Do not know", and "No response" were withdrawn because they were not relevant to the study.

**Table 2.** Industrial sectors.

| Industrial Sector | ID |
|---|---|
| Basic metals | B_Metals |
| Beverages | Beverages |
| Chemicals (exc. HPPC) | Chemicals |
| Coke and refined petroleum products | Coke_Petrol |
| Computers, electronics, and optical products | Electronics |
| Electrical equipment | Electrical |
| Food products | Food |
| Footwear and parts | Footwear |
| Furniture | Furniture |
| HPPC (Soap, detergents, and other cleaning preparations products) | HPPC |
| Leather and related products | Leather |
| Machinery and equipment | Machinery |
| Metal products (except Machinery and equipment) | Metal |
| Mining of metal ores | Min_Metals |
| Mining of non-metal ores | Min_Nmetals |
| Motor vehicles, trailers, and semi-trailers | Vehicles |
| Non-metallic mineral products | Non_Metals |
| Other manufacturing | Other_Mfg |
| Other transport equipment | Transport |
| Pharmaceutical chemicals and pharmaceuticals | Pharmaco |
| Plastics products | Plastics |
| Printing and reproduction of recorded media | Printing |
| Pulp and paper | Paper |
| Repair and installation | Repair |
| Rubber products | Rubber |
| Textiles products | Textile |
| Wearing apparel | W_Apparel |
| Wood products | Wood |

### 3.3. Step 3—Competitiveness Ranking

Two approaches were performed, one according to the opinion of the specialists of each sector and the second with industry in general. The MOORA method was used with two weight approaches for the criteria. The first approach treats the alternatives and criteria with different weights. In contrast, in the second approach, there is the integration of the Fuzzy Delphi method to unify the weights. MOORA approaches were applied to establish the ranking of the most competitive industrial sectors against the use of the digital technologies of Industry 4.0. The method chosen is an important factor for executing different types of information and problems [76]. MOORA was chosen since other multiple objective methods are criticized for the weighted linearity of different objectives [77]. The MOORA method is more powerful than other methods when considering its computational time, simplicity, mathematical calculations, stability, and types of information [78]. In this linear method, multiple objectives are replaced by a super-objective, prioritizing powerful alternative solutions, while an intermediate alternative does not rank first [77].

On the other hand, the need for the fuzzy theory arose because human judgments about preferences are always difficult to estimate from numerical values [79]. The main advantage of Fuzzy Delphi for group decision-making is that each expert opinion will be considered and integrated to reach a consensus on decision-making [33].

The MOORA method with changes in weight assignments for the criteria is detailed in Step 3.1. The MOORA method with Fuzzy Delphi for consensus of the attribution of weights in the criteria is described in Step 3.2.

### 3.3.1. Step 3.1—MOORA Method

According to [77], the MOORA method starts with a decision matrix (1) showing the performance of different alternatives (industrial sector) about the criteria (level of digital technologies utilization).

$$X_{ij} = \begin{bmatrix} X_{11} & X_{12} & \dots & \dots & \dots & X_{1n} \\ X_{21} & X_{22} & \dots & \dots & \dots & X_{2n} \\ \dots & \dots & \dots & \dots & \dots & \dots \\ X_{m1} & X_{m2} & \dots & \dots & \dots & X_{mn} \end{bmatrix} \tag{1}$$

where $X_{ij}$ is the measure of performance of the industrial sectors $j$ in given technology $i$, $m$ is the number of industrial sectors analyzed (alternatives), and $n$ is the number of technologies (criteria). Next, a relationship system is developed where each industrial sector is compared to the other profiles for each analyzed technology index. This results in a denominator of performance for the respective technology. This denominator is calculated by the square root of the sum of the squares of each industrial sector in each digital technology listed. This relation is expressed by Equation (2).

$$X_{ij} = \frac{X_{ij}}{\sqrt{\sum_{j=1}^{m} X_{ij}^2}} \tag{2}$$

where $X_{ij}$ is a dimensionless number that belongs to the interval [0, 1], representing the industrial profile $j$ in digital technology $i$; for multiobjective optimization, these normalized performances are added in maximization and subtracted in case of minimization. Thus, Equation (3) becomes:

$$Y_i = \sum_{i=1}^{i=g} w_i * X_{ij} - \sum_{i=g+1}^{i=n} w_i * X_{ij} \tag{3}$$

where $g$ is the utilization rate of the technologies to be maximized, (*ng*) is the technologies to be minimized, $Y_i$ is the normalized valuation value of the alternative concerning all the criteria, and $w_i$ is the weight for each criterion. In this study, all digital technology indexes were considered maximized. However, unlike the traditional MOORA method, in this step, the weights will be assigned individually for each criterion $i$ according to alternative $j$. For example, for companies in the plastics industry, only the index of importance that this industry profile has assigned to the 11 technologies will be used. That is, the opinion of the plastic sector respondents regarding the importance of technology does not interfere with any other type of industrial sector (alternative). In this method, each industry sector defined the importance of the digital technologies of Industry 4.0 for their respective business. Thus, the value of $w_i$ is replaced by $w_{ij}$, resulting in Equation (4):

$$Y_i = \sum_{i=1}^{i=g} w_{ij} * X_{ij} - \sum_{i=g+1}^{i=n} w_{ij} * X_{ij} \tag{4}$$

With the addition of this difference in weight assignment, considering the distinction between criteria and alternatives, the method is called w-MOORA.

### 3.3.2. Step 3.2—Fuzzy Delphi Method

The Fuzzy Delphi method consolidates the different response indexes of the industrial sectors and the importance of using digital technologies to boost competitiveness. In this way, the opinion of all the industrial sectors was considered to assign a single weight to the importance of each digital technology. The method was operationalized through Equation (1):

$$G_i = \frac{(U_i - L_i) + (M_i - L_i)}{3} + L_i \tag{5}$$

In this equation, $G_i$ is the consensus score among the experts, $U_i$ is the maximum value between the answers, $L_i$ is the minimum value between the answers, and $M_i$ is the geometric mean calculated from the expert opinions. However, in this study, no alternative was removed after finding the $G_i$ value. With the consensus value among the experts of the different industries, Equation (3) was changed, with Gi replacing the value of $w_i$, and Equation (6) was obtained.

$$Y_i = \sum_{i=1}^{i=g} G_i * X_{ij} - \sum_{i=g+1}^{i=n} w_i * X_{ij} \tag{6}$$

With this substitution in the attribution of weights through the integration of the Fuzzy Delphi method, this second ranking is called FD-MOORA.

*3.4. Data Analysis*

The first step of data analysis was to identify the weights of the digital technologies using our mathematical procedures. The w-MOORA method provided evidence of the adoption patterns of each industrial sector, while the FD-MOORA method provided an overview of the most relevant digital technologies. Then, we ranked the industrial sectors and compared the results from the two procedures using the $Y_i$ value concerning all criteria. We also verified the discrepancies in the results by plotting a graphic analyzing the two procedures (w-MOORA and FD-MOORA). All these steps were performed using Question (i) from the CNI survey. Afterward, we provided an overview of the implemented technologies in each industrial sector using questions (i) and (ii). Finally, we support our findings by presenting a framework illustrating the adoption patterns of Industry 4.0 for each industrial sector.

**4. Results**

We used the w-MOORA method to define the weights for the competitiveness of each Industry 4.0 digital technology using Question (i). We evaluated the scores of 11 digital technologies related to Industry 4.0 in 28 industrial sectors from the CNI survey [38]. Table 3 shows our categorizations of the considered variables in our analysis.

**Table 3.** Weight of each digital technology in the w-MOORA method.

| Sector | Digital Technologies | | | | | | | | | | |
|---|---|---|---|---|---|---|---|---|---|---|---|
| | CAD-CAM | IES | DAwS | DAS | Flex | MES-SCADA | Simulation | Big Data | IoT-PSS | 3D | Cloud |
| Min_Metals | 0 | 21 | 7 | 21 | 29 | 21 | 0 | 29 | 0 | 7 | 0 |
| Min_Nmetals | 7 | 21 | 9 | 11 | 9 | 13 | 7 | 7 | 10 | 3 | 9 |
| Food | 5 | 19 | 3 | 19 | 17 | 13 | 6 | 13 | 7 | 3 | 10 |
| Beverages | 8 | 22 | 4 | 16 | 12 | 16 | 2 | 12 | 6 | 2 | 10 |
| Textile | 8 | 26 | 4 | 17 | 25 | 10 | 3 | 12 | 12 | 6 | 13 |
| W_Apparel | 16 | 14 | 2 | 9 | 17 | 11 | 4 | 11 | 4 | 7 | 9 |
| Leather | 2 | 19 | 5 | 26 | 9 | 5 | 2 | 9 | 5 | 0 | 16 |
| Footwear | 17 | 27 | 2 | 20 | 17 | 10 | 2 | 10 | 2 | 12 | 10 |
| Wood | 14 | 23 | 3 | 15 | 16 | 11 | 1 | 14 | 8 | 4 | 1 |
| Paper | 3 | 21 | 5 | 34 | 18 | 13 | 3 | 12 | 13 | 8 | 7 |
| Printing | 7 | 8 | 3 | 18 | 16 | 9 | 3 | 13 | 17 | 15 | 18 |
| Coke_Petrol | 2 | 17 | 2 | 36 | 15 | 17 | 9 | 13 | 9 | 2 | 9 |
| Chemicals | 4 | 13 | 2 | 17 | 16 | 11 | 4 | 18 | 11 | 4 | 9 |
| HPPC | 5 | 19 | 3 | 24 | 30 | 8 | 11 | 8 | 16 | 11 | 16 |

**Table 3.** *Cont.*

| Sector | Digital Technologies | | | | | | | | | | |
|---|---|---|---|---|---|---|---|---|---|---|---|
| | CAD-CAM | IES | DAwS | DAS | Flex | MES-SCADA | Simulation | Big Data | IoT-PSS | 3D | Cloud |
| Pharmaco | 7 | 10 | 3 | 13 | 17 | 10 | 3 | 17 | 10 | 13 | 13 |
| Rubber | 11 | 11 | 6 | 14 | 14 | 19 | 3 | 8 | 3 | 11 | 6 |
| Plastics | 9 | 24 | 1 | 21 | 25 | 16 | 4 | 15 | 16 | 9 | 12 |
| Non_Metals | 6 | 21 | 3 | 19 | 20 | 10 | 2 | 9 | 11 | 5 | 6 |
| B_Metals | 7 | 25 | 6 | 20 | 26 | 15 | 6 | 22 | 9 | 10 | 13 |
| Metal | 17 | 33 | 4 | 14 | 12 | 7 | 6 | 11 | 11 | 8 | 10 |
| Electronics | 7 | 20 | 7 | 11 | 22 | 15 | 2 | 7 | 22 | 11 | 20 |
| Electrical | 11 | 33 | 2 | 27 | 24 | 16 | 15 | 7 | 15 | 13 | 13 |
| Machinery | 28 | 38 | 3 | 19 | 22 | 9 | 11 | 14 | 11 | 18 | 7 |
| Vehicles | 12 | 31 | 3 | 30 | 24 | 20 | 10 | 16 | 8 | 20 | 5 |
| Transport | 10 | 27 | 3 | 17 | 7 | 30 | 10 | 10 | 3 | 10 | 3 |
| Furniture | 18 | 34 | 2 | 15 | 17 | 8 | 6 | 8 | 10 | 9 | 9 |
| Other_Mfg | 15 | 21 | 0 | 18 | 18 | 8 | 5 | 8 | 15 | 23 | 13 |
| Repair | 13 | 31 | 0 | 9 | 9 | 6 | 3 | 16 | 6 | 3 | 6 |

We used the following criteria to determine the relevance of each technology: low ≤ 10 (not highlighted); medium = 11–18 (light gray highlighted); and high ≥ 19 (dark gray highlighted). As can be seen, most industrial sectors have at least one digital technology on a large scale (i.e., with the greatest potential to boost the competitiveness of the Brazilian industry over the next five years). Besides, most industrial sectors have two or more digital technologies on a medium scale. These findings show that the respondents expect good results in competitiveness from adopting Industry 4.0 digital technologies within five years. However, some digital technologies such as DAwS and Simulation are not expected to provide good results, with low scale as the main answer. The Fuzzy Delphi method helped us to obtain the weights for use in the FD-MOORA method. In other words, the opinion of all the industrial sectors was considered to assign a single weight to the importance of each digital technology. We used this procedure to understand the general importance level of digital technologies. Table 4 presents the overall ranking of the most relevant digital technologies.

**Table 4.** Weight of each digital technology in the FD-MOORA method.

| Digital Technology | Weight | % |
|---|---|---|
| IES | 22.36 | 17.0 |
| DAS | 20.94 | 15.9 |
| Flex | 17.96 | 13.6 |
| Big Data | 15.89 | 12.1 |
| MES-SCADA | 15.59 | 11.8 |
| CAD-CAM | 9.33 | 7.1 |
| 3D | 7.67 | 5.8 |
| IoT-PSS | 7.33 | 5.6 |
| Cloud | 6.67 | 5.1 |
| Simulation | 5.00 | 3.8 |
| DAwS | 3.00 | 2.3 |

The digital technologies considered with the greatest potential for increasing competitiveness were: IES (integrated engineering systems), DAS (digital automation with sensors), Flex (flexible lines), Big Data, and MES-SCADA. Technologies such as flexible lines and big data are considered some of the most disruptive in Industry 4.0 [39,46]. At the same time, MES-SCADA, IES, and DAS are called integrative digital technologies and support the integration and connection of all elements of the factory [40]. As evidenced, DAwS and Simulation have a low weight (lowest potential) when we incorporated FD-MOORA

with w-MOORA, corroborating the previous results. Based on these data, we elaborated on the ranking of the Brazilian industry sectors with the most potential for competitiveness utilizing Question (i). The results of the two rankings are presented in Table 5.

**Table 5.** Ranking of industrial sectors.

| Ranking | w-MOORA | | FD-MOORA | |
| | Sector | Yi | Sector | Yi |
| --- | --- | --- | --- | --- |
| 1 | Electrical | 55.1651 | Electronics | 40.7834 |
| 2 | Electronics | 49.8173 | Electrical | 40.5772 |
| 3 | Vehicles | 43.1807 | Plastics | 30.5929 |
| 4 | Machinery | 42.8381 | Vehicles | 29.8761 |
| 5 | Plastics | 35.3683 | B_Metals | 26.6010 |
| 6 | B_Metals | 30.9763 | Paper | 26.2447 |
| 7 | Coke_Petrol | 29.9994 | Textile | 26.1726 |
| 8 | Textile | 28.4796 | Machinery | 25.7714 |
| 9 | Other_Mfg | 27.0488 | Coke_Petrol | 25.0509 |
| 10 | Paper | 26.5999 | Chemicals | 23.9830 |
| 11 | Metal | 23.8004 | Other_Mfg | 22.4028 |
| 12 | Min_Metals | 23.2289 | Metal | 22.0490 |
| 13 | HPPC | 20.4258 | Min_Metals | 20.7159 |
| 14 | Chemicals | 20.3841 | Pharmaco | 20.5062 |
| 15 | Footwear | 20.2343 | Footwear | 19.1770 |
| 16 | Pharmaco | 19.8585 | Wood | 17.6642 |
| 17 | Furniture | 19.7492 | Food | 17.5584 |
| 18 | Printing | 16.5232 | Furniture | 17.5274 |
| 19 | Food | 16.2552 | Beverages | 16.3678 |
| 20 | Wood | 15.1437 | HPPC | 15.2053 |
| 21 | Beverages | 14.9040 | W_Apparel | 14.7414 |
| 22 | Leather | 13.8279 | Non_Metals | 14.3364 |
| 23 | W_Apparel | 13.1194 | Rubber | 14.2562 |
| 24 | Non_Metals | 13.0229 | Transport | 14.1023 |
| 25 | Rubber | 11.6315 | Leather | 13.8862 |
| 26 | Transport | 11.3127 | Min_Nmetals | 13.7942 |
| 27 | Min_Nmetals | 10.9498 | Printing | 13.2767 |
| 28 | Repair | 9.4295 | Repair | 10.2689 |
| | Average | 23.69 | Average | 21.20 |
| | Standard Deviation | 12.17 | Standard Deviation | 7.77 |

In the w-MOORA ranking, the five most competitive industrial sectors in Brazil regarding the use of digital technologies of Industry 4.0 are (1) electrical equipment, (2) computers, electronics, and optical products, (3) motor vehicles, trailers, and semi-trailers, (4) machinery and equipment, and (5) plastics products. These five industrial sectors had higher indexes, $Y_i > 35$. From the sixth position in this ranking, the final indexes have smaller differences, all under $Y_i < 31$. The five least competitive industrial sectors are (25) rubber products, (26) other transport equipment, (27) mining of non-metal ores, and (28) repair and installation, all under $Y_i < 12$. In the FD-MOORA ranking, the top five positions are (1) computers, electronics, and optical products, (2) electrical equipment, (3) plastics products, (4) motor vehicles, trailers, and semi-trailers, and (5) basic metals with $Y_i > 26.5$.

The rankings obtained by the two methods have differences and similarities. These differences and similarities are because, while the w-MOORA method considers only the specialists' view of the industrial sector to calculate their respective competitiveness, the FD-MOORA method covers the opinion of all specialists, converging on a single weight to each technology. When different opinions are considered, it is possible to identify that the range of the final ranking is higher, with a standard deviation of 12.17. On the other hand, utilizing a single weight, the competitiveness index between the sectors is more leveled

with a smaller amplitude, resulting in a standard deviation of 7.77. The ranking difference obtained between the two methods is detailed in Figure 2.

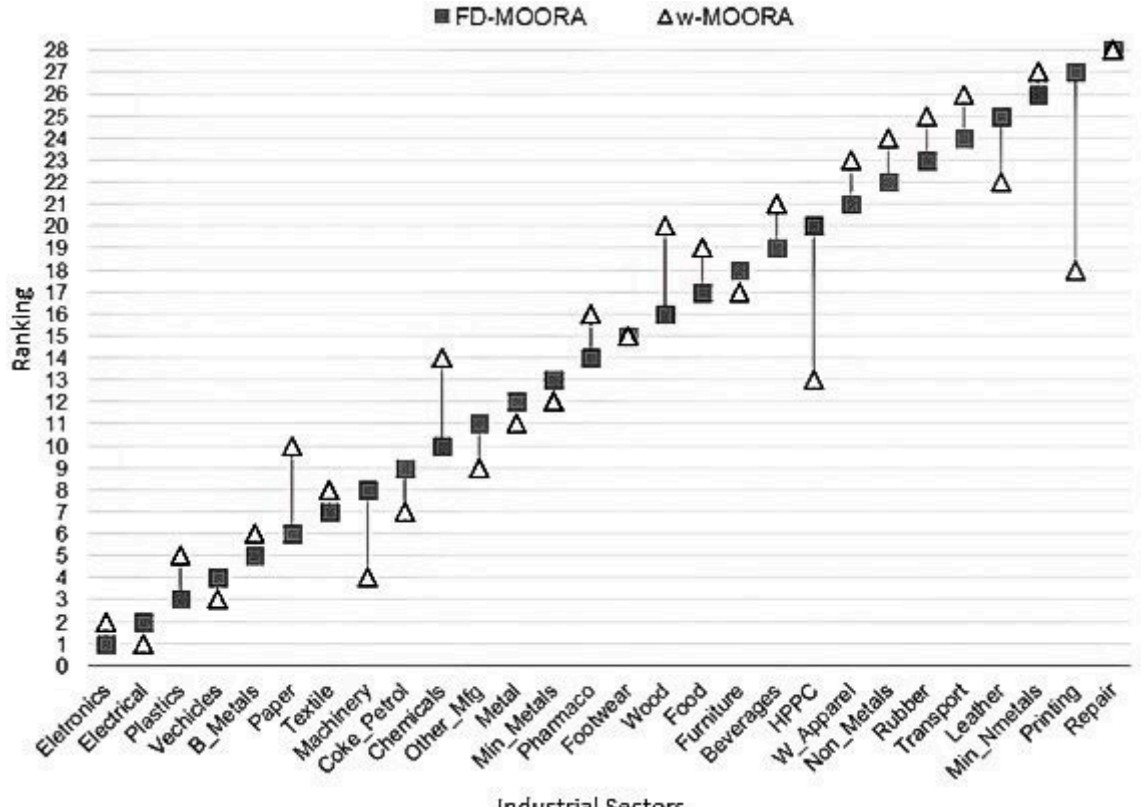

**Figure 2.** Ranking differences between w-MOORA and FD-MOORA.

It is observed that in both rankings, the first positions do not suffer excessive differences, showing that the opinion of the specialists of these companies converges with the general opinion of the Brazilian industry. The sharpest differences (above three positions) are in the sectors of pulp and paper (four positions), machinery and equipment (four positions), chemicals (four positions), and wood products (four positions), with the largest differences in the HPPC (nine positions) and printing and reproduction of recorded media (nine positions).

The companies that obtained a better position in the FD-MOORA ranking compared to w-MOORA are the companies in the pulp and paper, chemicals, and wood products industrial sectors. On the other hand, the industrial sectors that obtained a better position in the w-MOORA ranking and greater differences in comparison to the other ranking are machinery and equipment, HPPC, and printing and reproduction of recorded media. To summarize and complement the findings, Figure 3 represents the level of competitiveness of all analyzed sectors and the utilization rate of digital technologies.

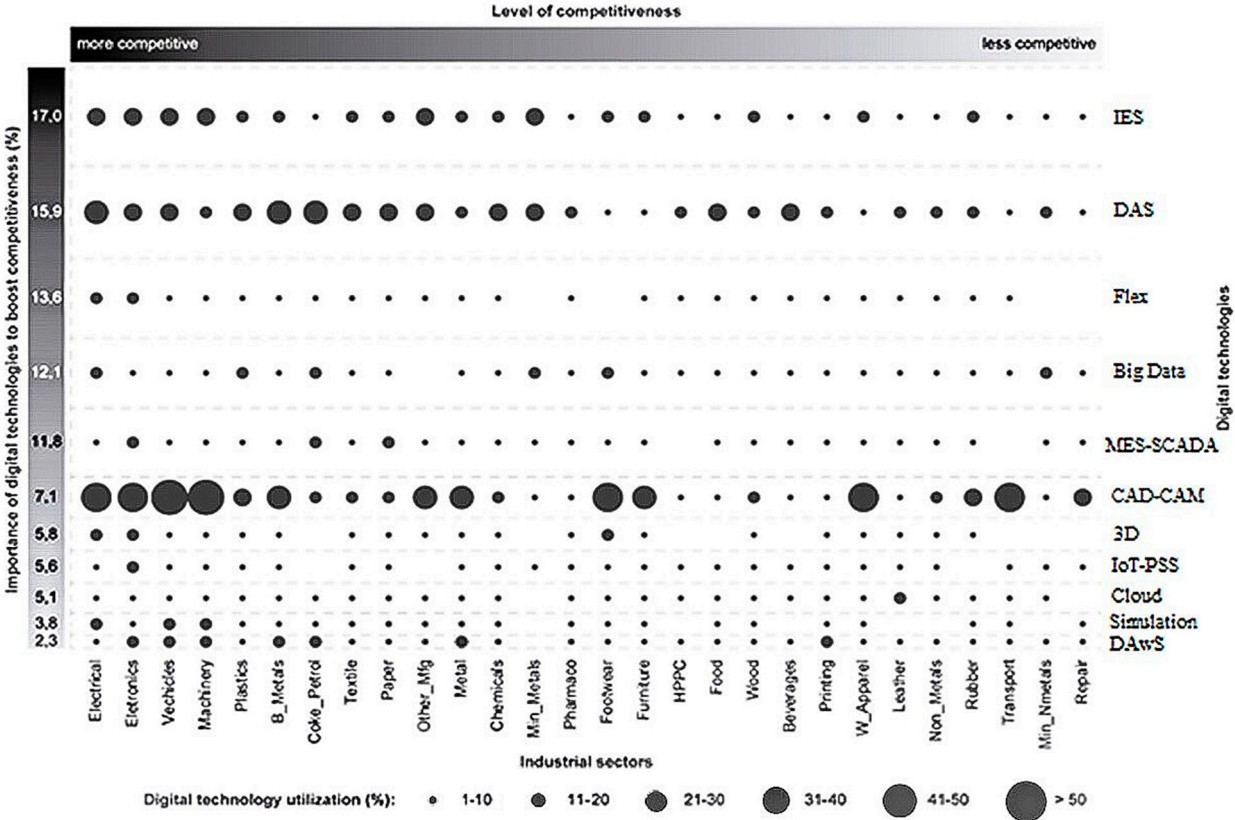

**Figure 3.** Level of competitiveness of the industrial sectors in Brazil in the Industry 4.0 scenario.

We used Question (i) from the CNI survey to illustrate which sectors of the Brazilian industry could achieve better competitiveness results using Industry 4.0 digital technologies. We also used Question (ii) to show what digital technologies are most utilized in each sector. The combination of Questions (i) and (ii) allowed us to understand the relationship between digital technologies already in use and the most prominent from the w-MOORA and FD-MOORA methods. The industrial sectors on the left are the most competitive, while those on the right are the least competitive. The circles represent the utilization rate of each digital technology for each industrial sector. On the left side, the important weight of each digital technology to boost competitiveness is represented. The range size illustrates the difference among weights. It is noted that the CAD-CAM, DAS, and IES technologies are the most utilized in Brazilian companies. The others do not have a high utilization percentage, with their thresholds below 21. All these analyses helped us to build Table 6, which illustrates the adoption patterns and digital technologies with high implementation in each industrial sector.

**Table 6.** Adoption patterns and digital technologies with high implementation per sector.

| Industrial Sector | Competitiveness Boost | | Already in Use |
|---|---|---|---|
| | **High** | **Medium** | **High** |
| Basic metals | [IES] [DAS] [Flex] [Big data] | [MES-SCADA] [Cloud] | [CAD-CAM] [DAS] |
| Beverages | [IES] | [DAS] [Flex] [MES-SCADA] [Big Data] | [DAS] |
| Chemicals (exc. HPPC) | | [IES] [DAS] [Flex] [MES-SCADA] [Big Data] [IoT-PSS] | [DAS] |
| Coke and refined petroleum | [DAS] | [IES] [Flex] [MES-SCADA] [Big Data] | [DAS] |
| Computers, electronics, and optical products | [IES] [Flex] [IoT-PSS] [Cloud] | [DAS] [MES-SCADA] [3D] | [CAD-CAM] [DAS] [IES] |

**Table 6.** *Cont.*

| Industrial Sector | Competitiveness Boost | | Already in Use |
|---|---|---|---|
| | **High** | **Medium** | **High** |
| Electrical equipment | [IES] [DAS] [Flex] | [CAD-CAM] [MES-SCADA] [Simulation] [IoT-PSS] [3D] [Cloud] | [CAD-CAM] [DAS] [IES] |
| Food products | [IES] [DAS] | [Flex] [MES-SCADA] [Big Data] | [DAS] |
| Footwear and parts | [IES] [DAS] | [CAD-CAM] [Flex] [3D] | [CAD-CAM] |
| Furniture | [IES] | [CAD-CAM] [DAS] [Flex] | [CAD-CAM] |
| HPPC (soap, detergents, and other cleaning preparations) | [IES] [DAS] [Flex] | [Simulation] [IoT-PSS] [3D] [Cloud] | |
| Leather and related | [IES] [DAS] | [Cloud] | |
| Machinery and equipment | [CAD-CAM] [IES] [DAS] [Flex] | [Simulation] [Big Data] [IoT-PSS] [3D] | [CAD-CAM] [IES] |
| Metal products (except Machinery and equipment) | [IES] | [CAD-CAM] [DAS] [Flex] [Big Data] [IoT-PSS] | [CAD-CAM] |
| Mining of metal ores | [IES] [DAS] [Flex] [MES-SCADA] [Big Data] | | [DAS] [IES] |
| Mining of non-metal ores | [IES] | [DAS] [MES-SCADA] | |
| Motor vehicles, trailers, and semi-trailers | [IES] [DAS] [Flex] [MES-SCADA] [3D] | [CAD-CAM] [Big Data] | [CAD-CAM] [DAS] [IES] |
| Non-metallic mineral products | [IES] [DAS] [Flex] | [IoT-PSS] | |
| Other manufacturing | [IES] [3D] | [CAD-CAM] [DAS] [Flex] [IoT-PSS] [Cloud] | [CAD-CAM] [DAS] [IES] |
| Other transport equipment | [IES] [MES-SCADA] | [DAS] | [CAD-CAM] |
| Pharmaceutical chemicals and pharmaceuticals | | [DAS] [Flex] [Big Data] [3D] [Cloud] | |
| Plastics products | [IES] [DAS] [Flex] | [MES-SCADA] [Big Data] [IoT-PSS] [Cloud] | [CAD-CAM] [DAS] |
| Printing and reproduction of recorded media | | [DAS] [Flex] [Big Data] [IoT-PSS] [3D] [Cloud] | |
| Pulp and paper | [IES] [DAS] | [Flex] [MES-SCADA] Big Data] [IoT-PSS] | [DAS] |
| Repair and installation | [IES] | [CAD-CAM] [Big Data] | [CAD-CAM] |
| Rubber products | [MES-SCADA] | [CAD-CAM] [IES] [DAS] [Flex] [3D] | [CAD-CAM] |
| Textiles products | [IES] [Flex] | [DAS] [Big Data] [IoT-PSS] [Cloud] | [DAS] |
| Wearing apparel | | [CAD-CAM] [IES] [Flex] [MES-SCADA] [Big Data] | [CAD-CAM] |
| Wood products | [IES] | [CAD-CAM] [DAS] [Flex] [MES-SCADA] [Big Data] | |

We used the same criteria for 'boost competitiveness' in the w-MOORA method. In the case of 'already in use', we used the criteria in Figure 3 for utilization percentage. Some digital technologies such as Big Data and DAwS have a considerable percentage in some industrial sectors, but we did not include them in Table 6 due to our criteria utilization. Next, we discuss these findings with our final framework.

## 5. Discussion

We summarized our findings in Figure 4, illustrating which industrial sectors align more with Industry 4.0 digital technologies. Moreover, we presented three levels for digital technologies, using the works [39,40] to support our discussion. Finally, we showed what digital technologies the five most prominent industrial sectors should invest in to achieve the Industry 4.0 level.

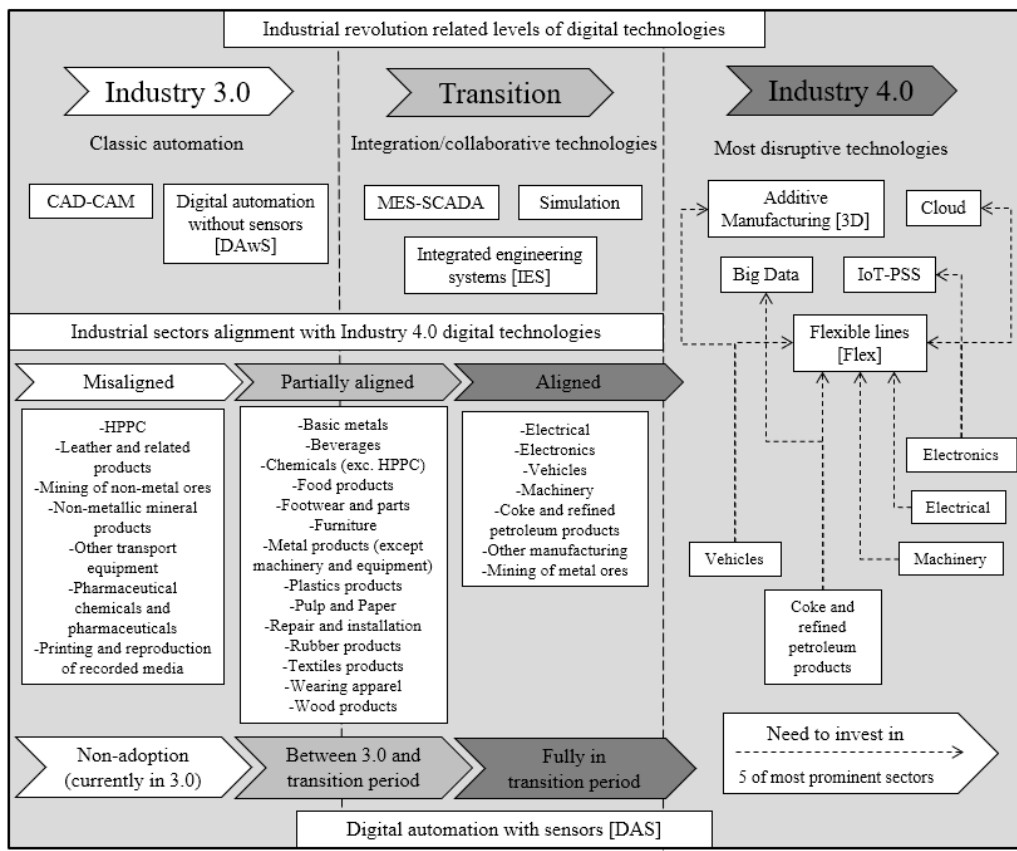

**Figure 4.** A framework of digital technologies for competitiveness.

We used this framework (Figure 4) with the support of Table 6 to guide discussions of our findings and clarify what digital technologies each industrial sector should invest in to boost its competitiveness. Sections 5.1 and 5.2 will answer the research questions (i) and (ii) presented at the end of Section 1.1.

*5.1. Current Status of Digital Technologies Implementation in Different Industry Sectors*

Firstly, we set up three levels for digital technologies towards Industry 4.0 using prior works as a basis [40–42]. The first level refers to consolidated digital technologies in the third industrial revolution. CAD-CAM systems and DAwS (digital automation without sensors) have been digital technologies since the third industrial revolution [38,41]. In other words, these digital technologies are in the early stages of classic automation. Nowadays, most industries use CAD-CAM systems in manufacturing and product planning through computerized systems [42,43]. Moreover, DAwS refers to automatized systems not using sensors in any industry with a minimum of automation architecture [38]. In the Brazilian industry, most of the sectors have CAD-CAM systems in their manufacturing and product planning, while DAwS has low adoption, but is still present.

The second level refers to integration and collaborative digital technologies, i.e., digital technologies which support the industries to create gateways in their manufacturing to achieve better results in industrial performance. At this level are MES-SCADA, IES, and Simulation technologies to support the industries in integrating and evaluating their internal processes. As stated, IES is considered the most important technology to boost competitiveness in the Brazilian industry. IES is considered the most important technology because most industrial sectors use it in their manufacturing processes. However, MES-SCADA technology supporting vertical integration [80] has low adoption in the Brazilian industry. This lack of adoption is a problem in achieving good results in industrial performance since MES-SCADA systems are technologies that have significant importance in achieving vertical integration in manufacturing [39]. Moreover, the industries have not

used their full potential despite simulation being present since the third industrial revolution, according to the Industry 4.0 related literature [81,82]. In other words, this technology refers to the simulation of virtual environments and their analysis. Ref. [83] states that most companies still do not have proper systems to analyze the data from the simulated environments. In the Brazilian industry, most industrial sectors do not use simulation as an analysis tool to improve their internal processes. In most cases, a simulation is a tool only for demand forecasting. This is a barrier for Brazilian industries since one of the goals of Industry 4.0 is the integration of the real with the virtual, making simulation a fundamental tool to achieving digital-twin systems [84].

The third level refers to the most disruptive technologies in Industry 4.0, i.e., integrated technologies that can greatly improve industrial performance [39]. At this level are Flex, Big Data, IoT-PSS, Cloud, and 3D technologies, to help the industries to achieve the Industry 4.0 stage. As can be perceived, all these technologies have low adoption in Brazilian industry. We could highlight the electronics and electrical sectors, which have a higher adoption level (but still low) of both 3D and Flex compared to other sectors. In the Industry 4.0 context, flexible lines and additive manufacturing are technologies that give a higher competitive advantage due to their countless opportunities for modularization and mass customization in the manufacturing processes and products [43,80]. As stated by [40], in an Industry 4.0 context, technologies such as big data and the cloud are integrated through IoT to enhance industrial performance. Therefore, most Brazilian industrial sectors do not have a high utilization percentage of big data and the cloud due to their difficulty integrating these technologies. In addition, in [39], the authors identified big data as negatively associated with the expected benefits for product performance. According to the authors, one possible explanation is that the Brazilian industry still does not perceive value in this technology for storage and analyzing data. This explains the low adoption of IoT-PSS in the Brazilian industry. IoT-PSS is a technology that incorporates product services through IoT platforms [85]. This technology is strongly related to product services through the connection of IoT and the use of big data and the cloud to enhance product performance. Since the Brazilian industries have low adoption of big data and the cloud, they will have low adoption of IoT-PSS technology due to the need to integrate these two technologies. Lastly, we understand that DAS is a technology that comprises the three levels presented in our framework. DAS has been present since the third industrial revolution and is a prerequisite to achieving Industry 4.0, being fundamental in the transition level [39,40].

These analyses and our previous results support us in understanding the alignment level of each Brazilian industrial sector with Industry 4.0. We found seven misaligned sectors: (1) HPPC (soap, detergents, and other cleaning preparations products; (2) leather and related products; (3) mining of non-metal ores; (4) non-metallic mineral products; (5) other transport equipment; (6) pharmaceutical chemicals and pharmaceuticals; and (7) printing and reproduction of recorded media. In other words, these sectors do not have a good implementation rate of any technology considered important to boosting their competitiveness. Therefore, these industrial sectors are considered the most misaligned with Industry 4.0 in Brazilian industry. None of these sectors are in the top ten in competitiveness in our two rankings using w-MOORA and FD-MOORA methods. Some results are surprising. For instance, pharmaceutical chemicals and pharmaceuticals is a sector that requires high technological infrastructure to develop medicine. Its misalignment with Industry 4.0 digital technologies concerns Brazilian industry, society, and government [38]. Other sectors such as HPPC and leather and related products are known in the Brazilian industry as industries with low technological architecture, lacking classic automation in their manufacturing process.

Although we illustrated three digital technology levels, we found that many industrial sectors are in a pre-transition period. In other words, industrial sectors are between Industry 3.0 and the transition period. We found 14 partially aligned sectors: (1) basic metals; (2) beverages; (3) chemicals (exc. HPPC); (4) food products; (5) footwear and parts; (6) furniture; (7) metal products (except machinery and equipment); (8) plastics products;

(9) pulp and paper; (10) repair and installation; (11) rubber products; (12) textiles products; (13) wearing apparel; and (14) wood products. These industrial sectors already incorporate digital technologies in their manufacturing process, but not at a high utilization rate. We can see that some industrial sectors (plastics, basic metals, textile and paper, and pulp) considered very competitive by the w-MOORA and FD-MOORA methods are present in this group. Although our mathematical procedures give these industrial sectors good prospects for competitiveness in the next five years, we can see that they still lack many digital technologies to be considered in the transition period. Table 6 in Section 4 presents the digital technologies that these industrial sectors should invest in in the next years to achieve the transition level and Industry 4.0. For instance, plastics and basic metals industrial sectors should invest more in IES and Flex technologies.

Concerning the most aligned industrial sectors towards Industry 4.0 direction, we found seven: (1) electrical; (2) electronics; (3) vehicles; (4) machinery; (5) coke and refined petroleum products; (6) other manufacturing; and (7) mining of metal ores. Most of these seven industrial sectors are ranked in the top 10 of w-MOORA and FD-MOORA methods. We showed that these sectors have a good utilization percentage in CAD-CAM, IES, and DAS technologies. Although none of the Brazilian industrial sectors are at an Industry 4.0 level, this group of industrial sectors is the most likely to achieve this competitive level in the following years. Furthermore, we noticed that these industrial sectors are in the transition period towards Industry 4.0, with a good adoption rate of fundamental digital technologies. However, as illustrated in Figure 4, these industrial sectors must invest in disruptive technologies to achieve this competitive level, as disruptive technologies have a low utilization rate in the Brazilian industry. Therefore, we selected five of our best-ranked industrial sectors from both mathematical procedures using the mean to indicate which digital technologies these sectors should invest in in the next years. According to our results, electronics and electrical are the most competitive and aligned sectors with Industry 4.0. Even though these two sectors have advantages over the others, they still lack investments in disruptive technologies in the Brazilian industry [38]. Finally, machinery and equipment, vehicles (automotive), and coke and refined petroleum products are characterized as industrial sectors with good utilization rates and a good competitiveness perspective. Despite these industrial sectors having good competitiveness perspectives in the following years, they still need more investments in disruptive technologies to foster the economic and technological development of the country.

*5.2. Framework of Digital Technologies in the Food Industry*

Brazilian industry is known as one of the largest exporters of food in the world and, therefore, investments in digital technologies for have vertical and horizontal integration are fundamental for the future, especially when thinking about productivity increases from the field to the industry, developing the concept of Agriculture 4.0 [37]. Figure 4 shows that the food industry is one of the industrial sectors that is partially aligned with the implementation of digital technologies of Industry 4.0 and is still in the process of knowledge and transition. Based on Table 6, Figure 5 was developed to illustrate the digital transformation process to boost food industry competitiveness towards Agriculture 4.0.

As stated in Figure 5, the food industry uses only DAS digital technology, i.e., sensors in food production and industry. However, using sensors alone may not bring great gains in competitiveness, especially if there are no other tools and technologies for control and integration. For DAS to be successful and for the food industry to boost competitiveness quickly, it is recommended to use DAS integrated with IES, giving rise to integrated engineering systems that can use the data obtained by the sensors to bring gains and improvements to the processes. This sensor data can come from primary production to processing by the industry. According to the results, for the food industry to enhance results in a consolidated manner and migrate to Agriculture 4.0, it should invest in a combined and simultaneous way in technologies such as IES, Flex, MES-SCADA, and Big Data. These combined technologies will ensure a process with control, traceability, and

feedback capability. Feedback is possible through big-data, where all decisions will feed back into the systems and serve as a basis for future decisions. This way, processes will no longer be based on experience, but on real data.

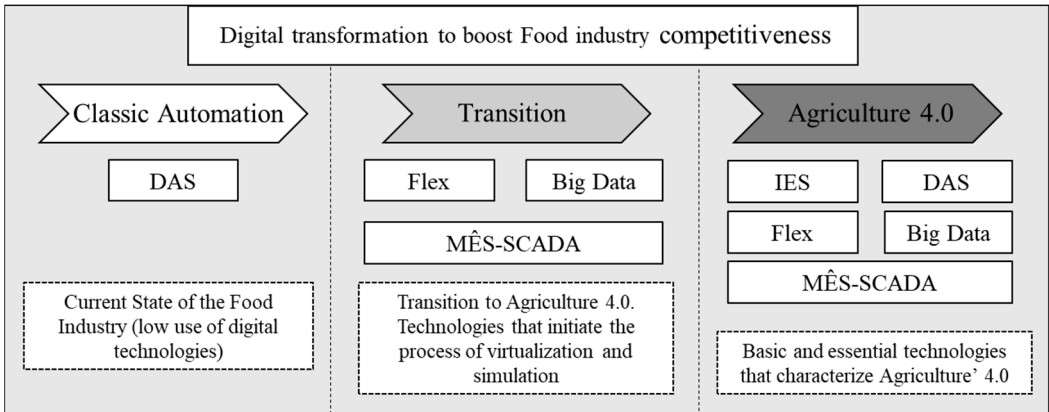

**Figure 5.** A framework of digital transformation to boost food industry competitiveness towards Agriculture 4.0.

In this way, Agriculture 4.0 can be explained by the digital transformation and use of digital technologies in the field, which allows crop growth, monitoring, and control of an irrigation system, and assists in choosing fertilizers [86]. On the flip side, Agriculture 4.0 and digital transformation in the food industry are changing how companies do business, establishing new relationships with customers, suppliers, and other stakeholders [51,53].

## 6. Conclusions

In this article, we analyze the level of competitiveness of 28 industrial sectors based on their perspectives on Industry 4.0 digital technologies. We evidenced that no sector of the Brazilian industry is engaged with disruptive technologies in their manufacturing processes. In addition, we defined three profiles (misaligned, partially aligned, and aligned) in the Brazilian industry, explaining their relationship with digital technologies in Industry 4.0. We discuss the reasons for these patterns, providing examples of behaviors in different industrial sectors in an emerging country, in this case Brazil. After defining these profiles, we further analyze the food industry, a key player in the global agro-industrial system. After the analysis and comparisons with other sectors, it was possible to identify that the food industry is partially aligned with digital technologies arising from advances in Industry 4.0 and Agriculture 4.0. This led us to conclude that the food industry needs attention to keep up with the technological development of other sectors to increase productivity and efficiency and enhance competitiveness.

The analysis of different industry sectors allowed us to visualize and indicate which technologies be leveraged for the competitiveness of the food industry, thus also taking a large step towards the development of the concept of Agriculture 4.0, which has the premise of applying technologies from primary food production to its processing. Agriculture 4.0, through technologies and digital transformation, can contribute to the sustainable increase of food production, especially in current times when the world population is growing rapidly and the adaptation of different crops and production systems to different demands must be faster. The industry is not engaged with disruptive technologies in its manufacturing processes.

The current literature has provided us with the most important digital technologies that are enablers of the concept of Agriculture 4.0. The present paper sought to investigate how the food industry can contribute to advancing the concept of Agriculture 4.0. To do so, we started with studies about the implementation of digital technologies in different industry sectors to understand the positioning of the food industry in this context. The results showed that the food industry has a low application of digital technologies and

that to improve its performance, some technologies are more suitable for developing the industry. The food industry is one of the links in the Agriculture 4.0 concept. It was found that research in this area is more recent than the research on Industry 4.0, so this paper sought to explore this gap. As a result, we present a framework composed of three phases; classical automation represents the first phase, that is, with the technologies currently used by the food industry. The second phase is the transition to Agriculture 4.0, indicating which digital technologies are fundamental to starting a digitalization process in the food industry. The third phase is Agriculture 4.0 because the basic and essential technologies that permeate this concept are already in place, ensuring better communication throughout the food production chain. The novelty of this article is highlighted in the framework of Figure 5. This figure can guide future research to enhance the transition of the food industry towards digitalization and implementation of technologies that make up Agriculture 4.0.

The investigation of several industry sectors was necessary to have an overview of the industry, finding patterns of technology adoption that could be shared across different sectors. In Figure 4, we present this overview based on CNI's database. Through Figure 4, it was possible to find adoption patterns for the food industry, which gave rise to Figure 5. Figure 5 can be very useful for the companies and producers that make up the food industry and can also serve as a basis for consultation by policymakers and universities. For the managers in the food industry, it serves as a basis for the diagnosis and preparation of companies to make investments in new technologies, bringing gains to production and processing and making the transition to Agriculture 4.0. For policymakers and universities, Figure 5 indicates which technologies would be more interesting to focus investments on, either through funding or research, because these are the technologies that most contribute to a rapid transition to Agriculture 4.0 according to the patterns studied in different sectors. In this sense, Figure 5 can serve as a guide for the digital transformation of the food industry in the transition to Agriculture 4.0.

*Limitations and Future Trends*

The technologies presented in the framework of Figure 5 are interconnected and allow the simulation of processes through virtual environments and their analysis. In emerging countries such as Brazil, the actors of the agroindustrial system still make little use of simulation and virtual systems as analysis tools to improve processes, improve productivity, or reduce production and processing costs. The identification of barriers and potential drivers for the application of these digital transformation technologies and making the transition to Agriculture 4.0 in Brazil's agroindustrial system can be the target of future research, since Brazil is one of the largest food producers in the world and can serve as a model for the application and development of these technologies in other countries. Another important area of research is to analyze the social and environmental impacts that the use of digital technologies in the industry brings to other links in the food production chain.

**Author Contributions:** Conceptualization, I.C.B., F.T.d.S., R.G.d.F.C., J.L.S., M.B.D.C., G.B.B. and E.O.B.N.; methodology, I.C.B., F.T.d.S., R.G.d.F.C., J.L.S., M.B.D.C., G.B.B. and E.O.B.N.; software, I.C.B., F.T.d.S., R.G.d.F.C., J.L.S., M.B.D.C., G.B.B. and E.O.B.N.; validation I.C.B., F.T.d.S., R.G.d.F.C., J.L.S., M.B.D.C., G.B.B. and E.O.B.N.; formal analysis, I.C.B., F.T.d.S., R.G.d.F.C., J.L.S., M.B.D.C., G.B.B. and E.O.B.N.; investigation, I.C.B., F.T.d.S., R.G.d.F.C., J.L.S., M.B.D.C., G.B.B. and E.O.B.N.; resources, I.C.B., F.T.d.S., R.G.d.F.C., J.L.S., M.B.D.C., G.B.B. and E.O.B.N.; data curation, I.C.B., F.T.d.S., R.G.d.F.C., J.L.S., M.B.D.C., G.B.B. and E.O.B.N.; writing—original draft preparation, I.C.B., F.T.d.S., R.G.d.F.C., J.L.S., M.B.D.C., G.B.B. and E.O.B.N.; writing—review and editing, I.C.B., F.T.d.S., R.G.d.F.C., J.L.S., M.B.D.C., G.B.B. and E.O.B.N.; visualization, I.C.B., F.T.d.S., R.G.d.F.C., J.L.S., M.B.D.C., G.B.B. and E.O.B.N.; supervision, I.C.B., F.T.d.S., R.G.d.F.C., J.L.S., M.B.D.C., G.B.B. and E.O.B.N.; project administration, I.C.B., F.T.d.S., R.G.d.F.C., J.L.S., M.B.D.C., G.B.B. and E.O.B.N.; funding acquisition, I.C.B. All authors have read and agreed to the published version of the manuscript.

**Funding:** This research was funded by CAPES—Brazil under the grant CAPES process Baierle No. 88887.464876/2019-00; and FAPERGS—Brazil under the grant FAPERGS process Baierle No. 22/2551-0000650-7.

**Institutional Review Board Statement:** Not applicable.

**Informed Consent Statement:** Not applicable.

**Data Availability Statement:** Not applicable.

**Conflicts of Interest:** The authors declare no conflict of interest.

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
