# Peer review of "Competitiveness of Food Industry in the Era of Digital Transformation towards Agriculture 4.0"

_sustainability, doi:10.3390/su141811779_

Round 1
Reviewer 1 Report
The author made an analysis of the different industrial sectors in order to investigate which technologies can leverage the competitiveness of the food industry. This research can serve as a guide. However, the following suggestions are recommended:
· Your abstract does not highlight the specifics of your research or findings. In introduction section I suggest: problems, Aim, Methods, Results, and Conclusion. The author needs to explain the major factors of the manuscript. The paper's aim has been evidenced very poor; it is strongly suggested to highlight the originality and added value of the present work with respect to the Literature about the same topic. Introduction suffers from a lack of motivation and innovations. It should be expanded to include a more detailed discussion of current problems.
· Contribution section is missing at the end of the introduction section. Add a contribution paragraph as a second last paragraph in the introduction section.
· Paper organization section is missing at the end of the introduction of section. Briefly describe the section and subsection of your whole menu script in one paragraph. Add this paragraph at the end of the introduction section.
· After the introduction section author should add a related work section
· Results and Discussion; the author should compare the finding of the present study with the previous study and justify for more clarity.
· Would you explicitly specify the novelty of your work? What progress against the most recent state-of-the-art similar studies was made?
· Conclusions should be amended to incorporate a broader discussion of the significance and potential application of this specific study.
· English throughout the manuscript needs to be improved.
· I would like to suggest this article for publication after minor changes. The article is not acceptable in its present form.
Author Response
Reviewer(s)' Comments to Author:
General Comments:
General Comment
The author made an analysis of the different industrial sectors in order to investigate which technologies can leverage the competitiveness of the food industry. This research can serve as a guide. However, the following suggestions are recommended:
Suggestions:
Suggestion 1 - S#1:
Your abstract does not highlight the specifics of your research or findings.
Answer to Suggestion 1 - S#1: Thanks for your comment. The abstract has been reformulated to explain the results of this research more directly. “The analysis of the different industrial sectors allowed us to present a framework of digital transformation to boost food industry competitiveness towards Agriculture 4.0. The results show that the food industry usually uses only one digital technology, showing the need for simultaneous and joint investments in the other technologies presented in this research. Public policies must be directed to encourage the expansion of the use of digital technologies in the food industry.
Suggestion 2 - S#2:
In the introduction section, I suggest: problems, Aim, Methods, Results, and Conclusion. The author needs to explain the major factors of the manuscript. The paper's aim has been evidenced very poor; it is strongly suggested to highlight the originality and added value of the present work with respect to the literature about the same topic. Introduction suffers from a lack of motivation and innovations. It should be expanded to include a more detailed discussion of current problems.
Answer to Suggestion 2 - S#2: Thanks for your suggestion. We followed his recommendations and rewrote the last three paragraphs of the Introduction. These paragraphs more clearly address the problem, objective, method, results, and conclusion. Now the results and conclusions could be better addressed in the final sections of the article.
Suggestion 3 - S#3:
Contribution section is missing at the end of the introduction section. Add a contribution paragraph as a second last paragraph in the introduction section.
Answer to Suggestion 3 - S#3: Thanks for your comment. The last three paragraphs of the Introduction have been rewritten to make the article's contributions more evident. The last paragraph of the Introduction brings more clearly the results and contributions. “The results show that the most developed industries regarding the application of digital technologies are the Electrical, Electronics, Plastics, and Vehicle industries, while the food industry is one of the industries that use digital technologies the least. Emerging countries can contribute a lot to developing digital technologies, as it does groundwork and preparation, which can be replicated in other countries. Finally, this article highlights which technologies are most commonly used in each industry sector and shows which technologies are the greatest drivers of competitiveness. This information should target future research, showing how it can be used in practice to become accessible to any type or size of the company. The advancement of research focused on certain technologies and applications contribute to the advancement of digital transformation in the food industry, also contributing to the development of the concept of Agriculture 4.0”.
This paragraph was taken up again in section 5, highlighting the results and contributions.
Suggestion 4 - S#4:
Paper organization section is missing at the end of the Introduction of section. Briefly describe the section and subsection of your whole menu script in one paragraph. Add this paragraph at the end of the introduction section.
Answer to Suggestion 4 - S#4: Thanks for your comment. The sentence was included at the end of Introduction section: “The remainder of the paper is organized as follows: section 2 details the methodological procedures used, section 3 presents the results obtained, section 4 brings a broad discussion about the findings, while section 5 presents the conclusions of the paper.”
Suggestion 5 - S#5:
After the introduction section author should add a related work section.
Answer to Suggestion 5 - S#5: Thanks for your comment. The section “2 – Correlated works” was created after the introduction section.
Suggestion 6 - S#6:
Results and Discussion; the author should compare the finding of the present study with the previous study and justify for more clarity.
Answer to Suggestion 6 - S#6: Thanks for your comment. We made it more evident that from reading the related papers, it was possible to present, in Figure 5, a framework of digital transformation to boost food industry competitiveness towards Agriculture 4.0. This framework was developed through the knowledge and evidence presented in the related papers. Based on this framework, the Conclusion section was reformulated, bringing new elements and making clearer the contribution of this research. Some elements not covered in this paper but also identified as research gaps have been reported in the Limitations and future research section. We thank you very much for this comment, which helped us see our contributions more clearly.
Suggestion 7 - S#7:
Would you explicitly specify the novelty of your work? What progress against the most recent state-of-the-art similar studies was made?
Answer to Suggestion 7 - S#7: Thanks for your comment. All your previous comments helped us to reorder our article and make more explicit elements such as problem, goal, results, and conclusion. The most relevant result is the framework in Figure 5. In conclusion, we included the following sentence to make a better closure to the article. “The current literature has provided us with the most important digital technologies and which of them are enablers for the concept of Agriculture 4.0. The present paper sought to investigate how the food industry can contribute to advancing the concept of Agriculture 4.0. To do so, we started with studies about the implementation of digital technologies in different industry sectors to understand the positioning of the food industry in this context. The results showed that the food industry has a low application of digital technologies and that to improve its performance, some technologies are more suitable for developing the industry. The food industry is one of the links in the Agriculture 4.0 concept. It was found that research in this area is more recent than the research on Industry 4.0, so this paper sought to explore this gap. As a result, we present a framework composed of three phases; the first phase is represented by the classical automation, that is, with the technologies currently used by the food industry. The second phase represents the transition to Agriculture 4.0, indicating which digital technologies are fundamental to starting a digitalization process in the food industry. In the third phase, we already have Agriculture 4.0 because the basic and essential technologies that permeate this concept are already in place and ensure better communication throughout the food production chain. The novelty of this article is highlighted in the framework of Figure 5. This figure can guide future research to enhance the transition of the food industry towards digitalization and implementation of technologies that make up the Agriculture 4.0.”
Suggestion 8 - S#8:
Conclusions should be amended to incorporate a broader discussion of the significance and potential application of this specific study.
Answer to Suggestion 8 - S#8: Thanks for your comment. We have amended the conclusion section according to your suggestion, this way, it was possible to explore and make more explicit the results and contributions of the research. The sentence was included in the Conclusion section: “The current literature has provided us with the most important digital technologies and which of them are enablers for the concept of Agriculture 4.0. The present paper sought to investigate how the food industry can contribute to advancing the concept of Agriculture 4.0. To do so, we started with studies about the implementation of digital technologies in different industry sectors to understand the positioning of the food industry in this context. The results showed that the food industry has a low application of digital technologies and that to improve its performance, some technologies are more suitable for developing the industry. The food industry is one of the links in the Agriculture 4.0 concept. It was found that research in this area is more recent than the research on Industry 4.0, so this paper sought to explore this gap. As a result, we present a framework composed of three phases; classical automation represents the first phase, with the technologies currently used by the food industry. The second phase represents the transition to Agriculture 4.0, indicating which digital technologies are fundamental to starting a digitalization process in the food industry. In the third phase, we already have Agriculture 4.0 because the basic and essential technologies that permeate this concept are already in place, ensuring better communication throughout the food production chain. The novelty of this article is highlighted in the framework of Figure 5. This figure can guide future research to enhance the transition of the food industry towards digitalization and implementation of technologies that make up the Agriculture 4.0.”
Suggestion 9 - S#9:
English throughout the manuscript needs to be improved.
Answer to Suggestion 9 - S#9: Thanks for your observation. The entire article has undergone an extensive English (American English) revision, including the inclusion and modification of sentences according to the reviewers' comments.
Suggestion 10 - S#10:
I would like to suggest this article for publication after minor changes. The article is not acceptable in its present form.
Answer to Suggestion 10 - S#10: Thanks for your appointment. All your suggestions were very important in improving the article and making it more attractive to readers.

Reviewer 2 Report
sustainability-1914742: This paper is well done and provides the valuable knowledge. However, a few perspectives need to be mentioned for the comprehensive discussion.
1) Figure 1 is not good resolution. Please improve.
2) Lie 194: The caption of Table 2 must be above the table, not below.
3) Line 283: What are the meaning for the dark gray highlighted and light gray highlighted in Table 3? Please explain the meaning of dark gray and light gray highlighted.
4) Line 297: What are the meaning for the dark gray highlighted in Table 3? Please provide the meaning.
5) Line 331: The name of x axis and y axis in Figure 2 must be added.
6) Figure 3 is not good resolution. Please improve.
7) Figure 3: What are the meaning of the difference size of dark circle in Figure 3? But the legends use blank circle symbol. Please improve.
8) Economic development and digital technology have to concern the sustainable environment. Production processes if industry and agriculture sectors generate water pollution, carbon emission, soil degradation, so on. Therefore, sustainable agriculture needs to pay attention. Please see these papers. [Carbon, Nitrogen and Water Footprints of Organic Rice and Conventional Rice Production over 4 Years of Cultivation: A Case Study in the Lower North of Thailand. Agronomy 2022, 12(2), 380.] [Strategies for reducing the carbon footprint of field crops for semiarid areas. A review. Agron. Sustain. Dev. 2011, 31, 643–656.]
9) Once economic and environment aspects are maintained, social (e.g. income, livelihood, …,) can be enhanced. Please discuss more details about this issue.
Author Response
Reviewer 2 Comments to Author:
General Comments:
General Comment
This paper is well done and provides the valuable knowledge. However, a few perspectives need to be mentioned for the comprehensive discussion.
Suggestions:
Suggestion 1 - S#1:
Figure 1 is not good resolution. Please improve.
Answer to Suggestion 1 - S#1: Thanks for your comment. Figure 1 was redone with a better resolution.
Suggestion 2 - S#2:
Line 194: The caption of Table 2 must be above the table, not below.
Answer to Suggestion 2 - S#2: Thanks for your observation. The caption was reordered above Table 2.
Suggestion 3 - S#3:
Line 283: What are the meaning for the dark gray highlighted and light gray highlighted in Table 3? Please explain the meaning of dark gray and light gray highlighted.
Answer to Suggestion 3 - S#3: Thanks for your comment. The meaning of the light and dark gray highlighted in Table 3 is to evidence the relevance of each technology: low ≤10 (not highlighted); medium=11-18 (light gray highlighted); and high≥19 (dark gray highlighted). The sentence “We used the following criteria to determine the relevance of each technology: low ≤10 (not highlighted); medium=11-18 (light gray highlighted); and high≥19 (dark gray highlighted)” was included after Table 3.
Suggestion 4 - S#4:
Line 297: What are the meaning for the dark gray highlighted in Table 3? Please provide the meaning.
Answer to Suggestion 4 - S#4: Thanks for your comment. The dark gray highlighted in Table 3 refers to each industrial sector's most relevant digital technology. To be more clear for readers, the sentence “We used the following criteria to determine the relevance of each technology: low ≤10 (not highlighted); medium=11-18 (light gray highlighted); and high≥19 (dark gray highlighted)” was included after Table 3. In the same way, Table 4 is an abstract of Table 3, and the dark gray highlighted refers to the most relevant digital technologies. So, the sentence “Table 4 presents the overall ranking of the most relevant digital technologies, highlighted in dark gray” was included before Table 4.
Suggestion 5 - S#5:
Line 331: The name of x axis and y axis in Figure 2 must be added.
Answer to Suggestion 5 - S#5: Thanks for your comment and observation. The name of the axis “x” and “y” was included in the new Figure 2
Suggestion 6 - S#6:
Figure 3 is not good resolution. Please improve.
Answer to Suggestion 6 - S#6: Thank you for your comment. We have redone Figure 3, but it again looks like a low resolution when we resized it to fit the journal template. We think the solution to this is to put it in landscape mode, but we will need permission from the Editorial Board when we are preparing the final document for publication.
Suggestion 7 - S#7:
Figure 3: What are the meaning of the difference size of dark circle in Figure 3? But the legends use blank circle symbol. Please improve.
Answer to Suggestion 7 - S#7: Thanks for your observation. We also put the blank circles' symbols in dark gray circles as the whole figure. The different size of circles refers to the level of competitiveness, as the legend indicates.
Suggestion 8 - S#8:
Economic development and digital technology have to concern the sustainable environment. Production processes if industry and agriculture sectors generate water pollution, carbon emission, soil degradation, so on. Therefore, sustainable agriculture needs to pay attention. Please see these papers. [Carbon, Nitrogen and Water Footprints of Organic Rice and Conventional Rice Production over 4 Years of Cultivation: A Case Study in the Lower North of Thailand. Agronomy 2022, 12(2), 380.] [Strategies for reducing the carbon footprint of field crops for semiarid areas. A review. Agron. Sustain. Dev. 2011, 31, 643–656.]
Answer to Suggestion 8 - S#8: Thanks for your comment. The section “2 – Correlated works” was created, and these papers about sustainable development were also included. This theme was cited again in the conclusion section, highlighted the importance of sustainability in the whole productive chain in which the food industry is inserted.
Suggestion 9 - S#9:
Once economic and environment aspects are maintained, social (e.g. income, livelihood, …,) can be enhanced. Please discuss more details about this issue.
Answer to Suggestion 9 - S#9: Thanks for your comment. In this article, the focus was not to study the economic, social, and environmental impacts, so we do not have enough elements to support this discussion. However, we agree that this is an extremely important and relevant issue because if the economic development of the food industry does not bring positive social and environmental impacts for the entire production chain, it becomes unattractive. Since we do not have elements to discuss this theme, we leave it as a suggestion for future research, addressing in more depth the social impacts that the use of digital technologies in the industry brings to other links in the food production chain.
“Another important research is to analyze the social and environmental impacts the use of digital technologies in the industry brings to other links in the food production chain.”

Reviewer 3 Report
The article presented for review presents the use of digital technology for the food industry in the context of Agriculture 4.0. Data collected refers to the industry in Brazil. The article is written clearly, the methodology was selected correctly. The results achieved have been discussed extensively in a separate chapter. I suggest minor adjustments: Line 161-163 - to be removed Table 2 - caption above the table Conclusions - Please change the description of the conclusions drawn from the results, not what this article brings.
Author Response
Reviewer 3 Comments to Author:
General Comments:
General Comment
The article presented for review presents the use of digital technology for the food industry in the context of Agriculture 4.0. Data collected refers to the industry in Brazil. The article is written clearly, the methodology was selected correctly. The results achieved have been discussed extensively in a separate chapter. I suggest minor adjustments:
Suggestions:
Suggestion 1 - S#1:
Line 161-163 - to be removed
Answer to Suggestion 1 - S#1: Thanks for your suggestion. In the present form, the sentence of lines 161-163 was removed. Another Reviewer asked to add a paragraph at the end of the introduction with the article organization sentence. So, I rewrote the initial sentence with the article's organization.
Suggestion 2 - S#2:
Table 2 - caption above the table
Answer to Suggestion 2 - S#2: Thanks for your observation. The caption was reordered above Table 2.
Suggestion 3 - S#3:
Conclusions - Please change the description of the conclusions drawn from the results, not what this article brings.
Answer to Suggestion 3 - S#3: Thanks for your comment. The most relevant result is the framework in Figure 5. We included the following sentence in conclusion, to make a better closure to the article. “The current literature has provided us with the most important digital technologies and which of them are enablers for the concept of Agriculture 4.0. The present paper sought to investigate how the food industry can contribute to advancing the concept of Agriculture 4.0. To do so, we started with studies about the implementation of digital technologies in different industry sectors to understand the positioning of the food industry in this context. The results showed that the food industry has a low application of digital technologies and that to improve its performance, some technologies are more suitable for the development of the industry. The food industry is one of the links in the Agriculture 4.0 concept. It was found that research in this area is more recent than the research on Industry 4.0, so this paper sought to explore this gap. As a result, we present a framework composed of three phases; the first phase is represented by the classical automation, that is, with the technologies currently used by the food industry. The second phase represents the transition to Agriculture 4.0, indicating which digital technologies are fundamental to starting a digitalization process in the food industry. In the third phase, we already have Agriculture 4.0 because the basic and essential technologies that permeate this concept are already in place and ensure better communication throughout the food production chain. The novelty of this article is highlighted in the framework of Figure 5. This figure can guide future research to enhance the transition of the food industry towards digitalization and implementation of technologies that make up the Agriculture 4.0.”

Round 2
Reviewer 2 Report
Accept in present form!
Author Response
Thanks for your comment. The entire article has undergone an extensive English (American English) revision, including the inclusion and modification of sentences according to the reviewers' comments.
